# Improvement of Enantiomeric L-Lactic Acid Production from Mixed Hexose-Pentose Sugars by Coculture of *Enterococcus mundtii* WX1 and *Lactobacillus rhamnosus* SCJ9

Augchararat Klongklaew [1], Kridsada Unban [2], Apinun Kanpiengjai [3], Pairote Wongputtisin [4], Punnita Pamueangmun [1], Kalidas Shetty [5] and Chartchai Khanongnuch [1,2,*]

[1] Interdisciplinary Program in Biotechnology, The Graduate School, Chiang Mai University, Chiang Mai 50200, Thailand; augchararat.taey@gmail.com (A.K.); pia_pannita39@hotmail.com (P.P.)
[2] Division of Biotechnology, School of Agro-Industry, Faculty of Agro-Industry, Chiang Mai University, Muang, Chiang Mai 50100, Thailand; kridsada_u@cmu.ac.th
[3] Division of Biochemistry and Biochemical Technology, Department of Chemistry, Faculty of Science, Chiang Mai University, Chiang Mai 50200, Thailand; apinun.k@cmu.ac.th
[4] Program in Biotechnology, Faculty of Science, Maejo University, Chiang Mai 50290, Thailand; pairotewong@gmail.com
[5] Department of Plant Sciences, Global Institute of Food Security and International Agriculture (GIFSIA), North Dakota State University, Fargo, ND 58108, USA; kalidas.shetty@ndsu.edu
* Correspondence: chartchai.k@cmu.ac.th; Tel.: +66-53-948-261

**Abstract:** Among 39 pentose-utilizing lactic acid bacteria (LAB) selected from acid-forming bacteria from the midgut of Eri silkworm, the isolate WX1 was selected with the highest capability to produce optically pure L-lactic acid (L-LA) from glucose, xylose and arabinose with furfural-tolerant properties. The isolate WX1 was identified as *Enterococcus mundtii* based on 16S rDNA sequence analysis. The conversion yields of L-LA from glucose and xylose by *E. mundtii* WX1 were 0.97 and 0.68 g/g substrate, respectively. Furthermore, L-LA production by *E. mundtii* WX1 in various glucose-xylose mixtures indicated glucose repression effect on xylose consumption. The coculture of *E. mundtii* WX1 and *Lactobacillus rhamnosus* SCJ9, a homofermentative LAB capable of producing L-LA from glucose clearly showed an improvement of L-LA production from 30 g/L total glucose-xylose (6:4). The results from Plackett–Burman design (PBD) indicated that Tween 80, $MnSO_4$ and yeast extract (YE) were three medium components that significantly influenced ($p < 0.05$) L-LA production using the coculture strategy in the presence of 2 g/L furfural. Optimal concentrations of these variables revealed by central composite design (CCD) and response surface methodology (RSM) were 20.61 g/L YE, 1.44 g/L Tween 80 and 1.27 g/L $MnSO_4$. Based on the optimized medium with 30 g/L total glucose-xylose (6:4), the maximum experimental L-LA value of 23.59 g/L reflecting 0.76 g/g substrate were achieved from 48 h fermentation at 37 °C. L-LA produced by coculture cultivated under standard MRS medium and new optimized conditions were 1.28 and 1.53 times higher than that obtained from single culture by *E. mundtii* WX1, respectively. This study provides the foundations for practical applications of coculture in bioconversion of lignocellulose particularly glucose-xylose-rich corn stover to L-LA.

**Keywords:** lactic acid bacteria; coculture; silkworm; L-lactic acid; medium optimization

## 1. Introduction

Lactic acid (LA) is used globally for various applications in food, pharmaceutical, cosmetic and chemical industries. Approximately 85% of LA is commonly used as an additive and a preservative in foods. In general, LA naturally exists in two optically active isomeric forms: L-LA and D-LA [1]. In particular, the increasing demand for optically pure LA is due to its potential use as a building block for synthesis of the biodegradable polymer named poly-LA (PLA) [2,3]. Both chemical synthesis and microbial bioconversion can be

used for LA production, but the chemical synthesis that is normally from petrochemical substrates is challenging due to racemic mixture of D- and L-LA, whereas the enantiomeric pure D- or L-LA can be achieved by the microbial fermentation [4]. The recent developments in industrial bioconversion technology and the advantage of microbial capability to selectively produce only D- or L-LA provides efficient strategies for industrial production of LA through microbial fermentation [5].

For microbial fermentation, the carbon source is the most important factor which directly impacts the production economics of LA fermentation. Among the simple fermentable sugars glucose and fructose are the most preferable sugars along with some waste byproducts containing disaccharides, such as lactose from whey and sucrose from molasses, which are conventionally used as important substrates for industrial LA production [6–8]. However, their applications are limited for LA production due to food security requirements for the use of such byproducts for food and feed bioconversions. Lignocellulosic materials as alternative sources of renewable sugars have currently gained more research interest than other carbon sources [9,10]. Lignocellulose is the most abundant biomass in the world and has the potential to create a new type of energy platform because of its availability and relatively low cost [11]. However, due to the complexity of lignocellulose structure and the difficulty associated with lignocellulose degradation, pretreatments are required to remove lignin, reduce cellulose crystallinity and increase the porosity of the materials. Hexose (6C) and pentose (5C) are two major components in carbohydrate component of lignocellulose [12] and pentose is typically found in hemicellulose, while hexose is mainly found in cellulose [13]. However, the main drawback of LA production from lignocellulosic substrate by most lactic acid bacteria (LAB) is the inability to consume pentose sugars, particularly xylose. Therefore, a large amount of pentose residues remains at the end of LA fermentative process. In addition, based on chemical hydrolysis of lignocellulosic materials, which are widely used as the carbon source in bio-refinery industry, hydroxymethyl furfural (HMF) and furfural are byproducts. The HMF and furfural amounts reported were in the range of 30–60 mg/g substrate depending on lignocellulosic materials [14]. Both HMF and furfural act as microbial growth inhibitors, thus limiting LA production from lignocellulosic hydrolysates [15]. To overcome the problem, microorganisms capable of producing LA from both hexose and pentose with furfural tolerant ability are required and are of relevance for further applications.

Based on the rationale described previously, LA-producing bacteria capable of using hexose and pentose were isolated from a novel source in the Eri silkworm (*Samia ricini*). This was a feasible target because the main diet of Eri silkworm for rearing is versatile lignocellulosic plants, which directly influence the carbon utilization of larval midgut microbial diversity. A previous study suggested that gut microbes and their enzymes are required for digestion of the nutritional substrates and can release fermentable sugars, amino acids and other molecules which are beneficial to the growth of the larva [16]. Therefore, Eri silkworm was targeted as a potential source for homofermentative LAB capable of pentose sugar consumption and also optically pure D- or L-LA production.

This specific study therefore describes the isolation and selection of the pentose-utilizing LAB with the capability for L-LA production and improvement of L-LA production from mixed hexose-pentose sugars by coculture fermentation. This mixed simple sugar study then provides foundation for advancing LA fermentation-based production from corn stover in future studies. Modification of nutritional components for higher LA yield using the statistical experimental designs is also described.

## 2. Materials and Methods

### 2.1. Isolation and Screening of Pentose-Utilizing LAB

Ten mature stage Eri silkworms were collected from local farm in Sankampang district, Chiang Mai province, Thailand. Eri silkworms were rinsed twice with sterile water. Then, cleaned worms were soaked in 70% (*v/v*) ethanol for 30 s, followed by a final rinse with sterile water. The gut tissue was then aseptically dissected from each worm and homog-

enized. Serial 10-fold dilution was performed by transferring 0.5 g of the homogenized sample into 4.5 mL 0.85% ($w/v$) NaCl, mixed vigorously, and 100 μL of each dilution was subjected to spread-plating on nutrient agar (NA) supplemented with 10 g/L glucose and 0.04% ($w/v$) bromocresol purple. After incubation at 37 °C for 24 h under anaerobic condition, the acid-forming bacterial isolates observed from the yellow clear zone surrounding the colony were replicated by transferring into NA containing 10 g/L xylose and 0.04% ($w/v$) bromocresol purple. Total viable cells were enumerated as colony forming units (CFU) per gram of each sample. The number of acid-forming colonies visualized by the yellow halo surrounding the colony on NA glucose and NA xylose were counted and calculated for the ratio of acid-forming bacterial isolates to the total viable bacterial number. A single colony of bacterial isolate capable of growth and producing acid on NA xylose were transferred into 10 mL de Man, Rogosa and Sharpe (MRS) broth (5 g/L yeast extract (YE), 10 g/L beef extract, 10 g/L peptone, 5 g/L sodium acetate, 1 g/L Tween 80, 2 g/L $K_2HPO_4$, 2 g/L tri-ammonium citrate, 0.2 g/L $MgSO_4 \cdot 7H_2O$, 0.2 g/L $MnSO_4 \cdot H_2O$) supplemented with 10 g/L xylose and 0.04% ($w/v$) bromocresol purple and incubated in static condition at 37 °C for 48 h. The isolate showing the growth and the yellow color was assumed to be LAB with LA-producing capability. The selected isolates were maintained in the commercial MRS broth (HiMedia, Mumbai, India) with 30% ($v/v$) glycerol at −80 °C for further investigation.

### 2.2. Selection of LAB for Optically Pure L-LA Production

The 39 isolates of LAB capable of producing LA were investigated to identify the optically pure isomeric forms produced using three different types of carbon sources including glucose, xylose and arabinose. One percent by volume of seed inoculum prepared from a single colony of each isolate was transferred to MRS broth with 10 g/L of glucose or xylose or arabinose as a sole carbon source and cultivated at 37 °C for 48 h with pH controlled at 7.0 manually using 10 N NaOH. After incubation, culture broth was centrifuged at 12,000× $g$ for 10 min at 4 °C and the supernatant was used to determine LA concentrations. The type of LA (D and L form) was determined enzymatically based on colorimetric method by measuring the absorbance at a wavelength of 340 nm as described in the manual of the enzymatic assay kit (K-DLATE, Megazyme International Ireland, Bray, Co. Wicklow, Ireland) following the manufacturers' instructions. The isolates showing the highest optically pure L-LA production based on xylose substrate were preliminary selected for further study.

### 2.3. Furfural Tolerance and Inhibitory Effect on L-LA Production

All 12 selected LAB isolates were tested for their furfural tolerance. Seed inoculum 1% ($v/v$) of each selected isolate was transferred into 20 mL MRS broth containing 10 g/L glucose supplemented with furfural in the range of 1 to 6 g/L, followed by incubation under the static cultivation at 37 °C for 48 h. The culture broth was separated by centrifugation at 12,000× $g$ for 10 min at 4 °C and L-LA was determined using the enzymatic assay kit as described previously. A similar experiment was also carried out using 10 g/L of xylose or arabinose instead of glucose.

### 2.4. Identification of LAB by 16S rRNA Gene Sequence Analysis

The molecular identification of LAB isolates was performed using the method described by Unban, et al. [17]. Briefly, 10 mL of bacterial culture broth cultivated overnight at 37 °C were centrifuged at 12,000× $g$ for 5 min. The bacterial cells were collected for genomic DNA extraction using the standard protocol as described by Sambrook and Russell [18]. Amplification of 16S rRNA gene was performed using genomic DNA as a template with primers 27F (5′-AGAGTTTGATCMTGGCTCAG-3′) and 1525R (5′-AAGGAGGTGWTCCARCC-3′) [19]. All reactions of PCR were performed using Phusion® High-Fidelity PCR Master Mix (New England Biolabs, Beverly, MA, USA). The PCR products were submitted for nucleotide sequencing service at 1st BASE Laboratory Company,

Malaysia. The 16S rRNA gene sequences were compared to other genes from NCBI Gen-Bank database and multiple sequence alignment were performed using BioEdit 7.0 software tool. The phylogenetic tree was constructed based on the neighbor-joining methods by MEGA version 4.0 software [20].

### 2.5. Effect of Different Glucose/Xylose Ratio on L-LA Production by Enterococcus mundtii WX1

Seed inoculum was prepared by transferring a single colony of *E. mundtii* WX1 into 20 mL commercial MRS broth and incubated at 37 °C under the static condition for 12 h for achieving OD600 of 0.6–0.8. Then, 1% (*v/v*) seed inoculum was transferred into 50 mL MRS medium with different ratio of the glucose and xylose including G20 (20 g/L glucose), G15 × 5 (15 g/L glucose and 5 g/L xylose), G10 × 10 (10 g/L glucose and 10 g/L xylose), G5X15 (5 g/L glucose and 15 g/L xylose), and X20 (20 g/L xylose), then cultivated at 37 °C for 48 h under static condition with pH controlled manually at 7.0 during fermentation. Samples were taken at 12 h-intervals for determination of LA, glucose and xylose using enzymatic test kits (K-DLATE, K-GLUC and K-XYLOSE, Megazyme International Ireland, Bray, Co. Wicklow, Ireland).

### 2.6. L-LA Production from Mixed Glucose and Xylose by Coculture Fermentation

*E. mundtii* WX1 was designed for coculture fermentation when combined with *Lactobacillus rhamnosus* SCJ9, a highly active homofermentative LAB capable of producing L-LA from only glucose [21]. The mixed sugar of glucose and xylose in a ratio of 6:4 was selected as a model for LA fermentation. Total of 1% (*v/v*) inoculum of *E. mundtii* WX1 together with 1% (*v/v*) inoculum of *L. rhamnosus* SCJ9 were transferred into MRS medium containing 12 g/L glucose and 8 g/L xylose, while, 2% (*v/v*) seed inoculum of *E. mundtii* WX1 was transferred to the same medium and used as the control. Then, both were cultivated at 37 °C for 48 h under static condition with pH at 7.0 manually controlled during fermentation. Samples were taken at 12 h intervals for 48 h in order to determine LA, glucose, xylose and viable cells.

### 2.7. Statistical Medium Optimization for L-LA Production by Coculture Fermentation

Plackett–Burman design (PBD) was applied to find the relative significance of nine variables influencing L-LA production by co-fermentation strategy combining *E. mundtii* WX1 and *L. rhamnosus* SCJ9. This experimental design was based on the MRS medium by using two-level independent variables including yeast extract, peptone, beef extract, Tween 80, tri-ammonium citrate, $MgSO_4 \cdot 7H_2O$, $K_2HPO_4$, $MnSO_4 \cdot H_2O$ and sodium acetate (Supplementary Materials, Table S1) with a fixed concentration of 30 g/L mixed carbon source (60% glucose: 40% xylose) based on what could be similar if complete hydrolysis of 100 g glucose-xylose rich corn stover [22] is used in future studies. LA production was optimized based on design matrix of PBD. The experimental LA was analyzed by analysis of variance (ANOVA) and regression analysis in order to fit the LA data with the first-order model according the following Equation (1);

$$Y = \beta_0 + \sum_{i=1}^{n} \beta_i x_i \tag{1}$$

where "Y" is the response factor, $\beta_0$ is the model coefficient and $\beta_i$ is the linear coefficient, $x_i$ are the independent variables and n is the number of independent variables.

Central composite design (CCD) was used to optimize the level and interaction of the most significant variables (Tween 80, $MnSO_4 \cdot H_2O$ and YE) identified by PBD. The design matrix was set with five levels of each variable including $-\alpha$, $-1$, $0$, $+1$, $+\alpha$ where $\alpha = 2^{k/4}$, and k is the number of independent variables. For statistical calculation, the actual values were statistically calculated by the following Equation (2):

$$x_i = \frac{x_i - x_0}{\Delta x_i} \ i = 1, 2, 3, \ldots, k \tag{2}$$

where $x_i$ is actual value of independent variables; $x_0$ is actual value of independent variables at center point; $\Delta x_i$ is the step change value. LA production was conducted based on the design matrix of PBD. The experimental LA was analyzed by ANOVA and regression analysis in order to fit the LA data with the second-order model according to the Equation (3):

$$Y = \beta_0 + \sum \beta_i x_i + \sum \beta_{ii} x_i^2 + \sum \beta_{ij} x_i x_j \tag{3}$$

where Y is the predicted response factor, $\beta_0$ is the constant coefficient, $\beta_i$ are linear coefficients, $\beta_{ij}$ are the second order interaction coefficients, $\beta_{ii}$ are the quadratic coefficients of the model and $x_i$, $x_j$ represents the independent factors in the terms of actual values [23]. After a reliable model was fitted to the second-order model, the predicted levels of the previous significant variables were obtained and used to predict the highest L-LA production. The predicted values were applied, and more than 90% validation was satisfactorily accepted.

### 2.8. Comparative Study on L-LA Production by the Optimized Medium

Seed culture of *E. mundtii* WX1 and *L. rhamnosus* SCJ9 were prepared by transferring the single colony of each bacterium to 20 mL commercial MRS broth and incubated at 37 °C for 24 h. A total of 1% (*v/v*) of each strain was transferred to 100 mL optimized medium containing 30 g/L mixed glucose-xylose (6:4) supplemented with 2 g/L furfural and incubated under static cultivation at 37 °C for 48 h. The aliquot of 3 mL culture broth was taken at 24 and 48 h for analysis of L-LA. Bacterial growth was also determined by drop plate technique and expressed in terms of viable cells. The experiment was compared with two different fermentation strategies: coculture of *E. mundtii* WX1 and *L. rhamnosus* SCJ9, and single culture of *E. mundtii* WX1 in 100 mL. The experiments were carried out in triplicate.

### 2.9. Statical Analysis

The differences between mean values in each experimental data were analyzed using one-way ANOVA and Tukey's multiple comparison test by the Statistix software version 8.0 (Analytical Software, Tallahassee, FL, USA) with significance defined as *p*-values less than 0.05. The statistical analysis of the PBD and CCD results was performed using the Design Expert software version 8.0 (Stat-Ease Inc., and Minneapolis, MN, USA). The program was also used to design the experiments as well as for the regression and graphical analysis of the experimental data obtained.

## 3. Results and Discussion

### 3.1. Bacterial Isolation and Screening of Xylose-Utilizing LAB

The observation of bacterial growth on NA supplemented with 1% (*w/v*) glucose for 24 h cultivation revealed bacterial population of $3.95 \times 10^7$ CFU/g, while the acid-forming bacteria observed from the yellow clear zone surrounding their colonies were found up to $3.76 \times 10^7$ CFU/g, which was 95.18% of the total bacterial counts. However, after transferring the acid-forming bacteria into NA supplemented with 1% (*w/v*) xylose, only $2.52 \times 10^7$ CFU/g was observed which represented 63.8% of total number of acid-forming bacteria. In total, 504 colonies of acid-forming bacteria capable of growth on NA xylose were randomly screened by transferring each colony into MRS broth containing xylose as the sole carbon source. After the secondary screening, 51 bacterial isolates (which represented 10.12% of the selected acid-forming bacteria) showed the ability to grow and produce acid, thus presumptively classified as LAB. Of these bacterial numbers, 39 isolates showed rapid growth and acid production within 12 h of cultivation in MRS broth containing xylose and their L-LA concentrations at 48 h are presented in Figure 1. The results from this experiment demonstrated the relevance of the microbial population of the Eri silkworm gut and further the abundance of acid-forming bacteria. Furthermore, the target LAB was found to be only 51 isolates of presumptive LAB which is only 10.12% of 504 isolates. This means approximately 90% or the majority of xylose-utilizing bacteria with acid-producing ability in the gut of Eri silkworm are non-LAB and the number of LAB found from this experiment is approximately 7% of the total number of culturable bacteria. This result

corresponded with the previous study which concluded that the dominant bacteria in gut of Eri silkworm belongs to the order Entrobacteriales (60%) followed by Bacillales (15%) and the majority of the culturable bacteria in gut of Eri silkworm were *Bacillus* spp., *Citrobacter* sp., *Enterobacter* sp., and *Pseudomonas* spp. [24]. LAB belongs to phylum Firmicutes which can be assigned in Bacilli class which includes Lactobacillales order. A previous study found that approximately 54% of aerobic culturable bacteria in Eri silkmoth, *Samia ricini* were in the phylum Firmicutes. Of these, 48 (representing 75%) of the bacteria produced cellulolytic, xylanolytic and lipolytic enzymes, and nitrate reductase, which are associated with food digestion and nutritional provision from digested foods [24]. Hence, it can be implied that Eri silkworm could be an alternative source of xylose-fermenting bacteria, specifically xylose-fermenting LAB based on the initial assumption of this study.

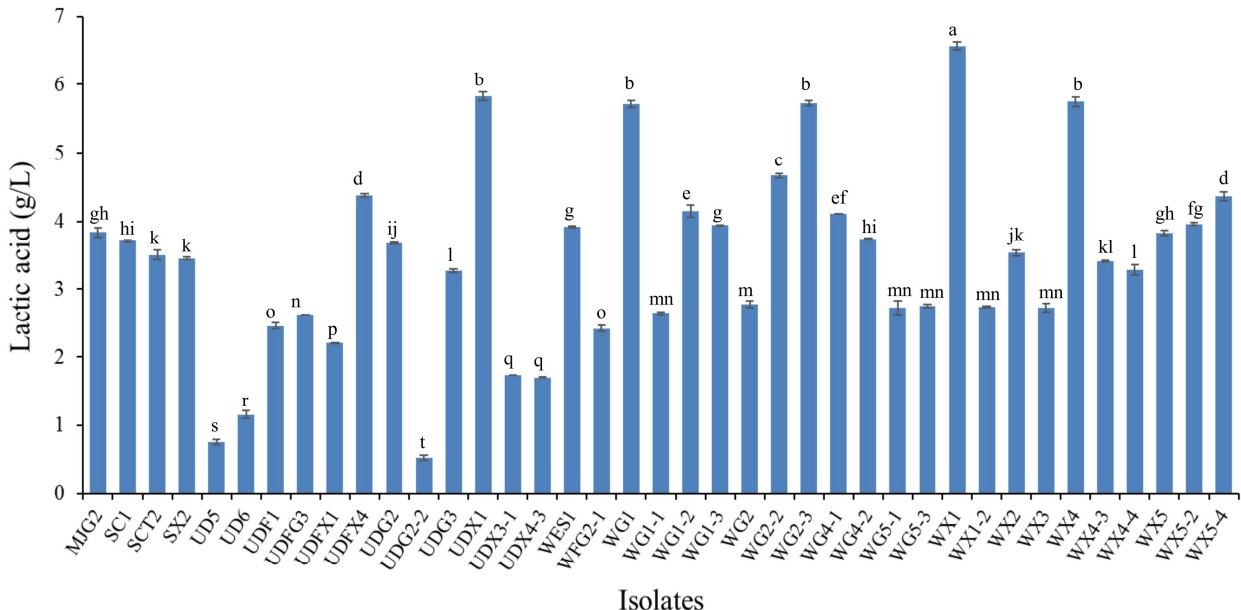

**Figure 1.** Lactic acid production from 39 isolates in MRS with 10 g/L xylose after cultivation at 37 °C for 48 h. Different small letters indicate significant differences between lactic acid produced from each isolate ($p < 0.05$).

### 3.2. Screening of LAB for Optically Pure L-LA Production

Considering the basic fermentable sugar composition obtained from chemical hydrolysis of lignocellulosic material, the previously selected 39 LAB isolates were evaluated for their fermentation potency using different single pure sugars including glucose, xylose and arabinose, which normally is found as the main sugar component in lignocellulosic material. It was found that 12 out of 39 LAB isolates were able to grow and produce high optically pure L-LA from glucose, xylose and arabinose and the L-LA purity ranged from 94–100% (Table 1) and LA content and its L-optical purity were varied depending on LAB strains. Two isolates, SCT2 and WX1, produced L-LA at the highest level in MRS glucose, but the isolate SCT2 produced L-LA from xylose and arabinose at levels of 3.20 and 1.15 g/L, respectively, while L-LA produced by the isolate WX1 were 6.80 and 6.00 g/L, respectively. Therefore, SCT2 was not suitable for further application in L-LA production from lignocellulose which is normally composed of hexose and pentose in various ratios. There were only five isolates including UDX1, WG1, WG2–3, WX1 and WX4 which produced L-LA at higher than 5 g/L in MRS broth containing 10 g/L xylose. Meanwhile, other isolates showed their xylose and arabinose fermenting ability at levels less than 3 g/L LA from 10 g/L of each carbon source. Yet, it is of high relevance to select LAB using simple sugar combination screening for further use in L-LA production from actual lignocellulosic materials which is composed of pentose as almost half of carbohydrate content. However,

their furfural tolerance should also be investigated further for application as this criterion is also important for LAB selection procedure.

**Table 1.** Enantiomeric purity of L-lactic acid production from glucose, xylose and arabinose at 10 g/L concentration in MRS broth by twelve selected presumptive LAB under static condition at 37 °C for 48 h.

| Isolates | Optical Pure L-Lactic Acid (%) | | | L-Lactic Acid Production (g/L) at 48 h | | |
|---|---|---|---|---|---|---|
| | MRS Glucose | MRS Xylose | MRS Arabinose | MRS Glucose | MRS Xylose | MRS Arabinose |
| MJG2 | 100 | 100 | 100 | 7.75 ± 0.02 [f] | 3.40 ± 0.04 [e] | 1.38 ± 0.01 [i] |
| SC1 | 96.3 | 95.9 | 94.5 | 8.11 ± 0.12 [e] | 3.37 ± 0.12 [e] | 1.15 ± 0.02 [j] |
| SCT2 | 97.2 | 97.6 | 94.3 | 9.41 ± 0.10 [a] | 3.20 ± 0.05 [e] | 1.15 ± 0.02 [j] |
| SX2 | 95.2 | 97.1 | 96.7 | 8.65 ± 0.03 [c] | 3.20 ± 0.06 [e] | 1.92 ± 0.01 [g] |
| UDX1 | 100 | 100 | 100 | 7.31 ± 0.05 [g] | 6.41 ± 0.12 [b] | 5.86 ± 0.03 [b] |
| WES1 | 100 | 100 | 100 | 8.37 ± 0.03 [d] | 3.68 ± 0.09 [d] | 1.78 ± 0.01 [h] |
| WG1 | 100 | 100 | 100 | 8.97 ± 0.11 [b] | 6.47 ± 0.10 [b] | 4.68 ± 0.04 [c] |
| WG2-3 | 100 | 100 | 100 | 7.40 ± 0.05 [g] | 6.54 ± 0.09 [b] | 4.20 ± 0.04 [d] |
| WX1 | 100 | 100 | 100 | 9.27 ± 0.13 [a] | 6.80 ± 0.05 [a] | 6.00 ± 0.07 [a] |
| WX2 | 100 | 100 | 100 | 8.98 ± 0.11 [b] | 3.36 ± 0.02 [e] | 2.56 ± 0.02 [f] |
| WX3 | 100 | 100 | 100 | 8.93 ± 0.12 [b] | 2.61 ± 0.01 [f] | 2.53 ± 0.01 [f] |
| WX4 | 100 | 100 | 100 | 8.62 ± 0.06 [cd] | 5.40 ± 0.05 [c] | 3.49 ± 0.02 [e] |

Note: Means in column with different superscripts are statistically different at $p < 0.05$.

### 3.3. Furfural Tolerance and Inhibitory Effect on L-LA Production

In the production process of LA from corn stover, chemical hydrolysis generated the main fermentable sugar including glucose and xylose, respectively in addition to releasing of furfural compounds [25]. In corn stover, glycan and xylan contents were 34.4% and 22.8% by weight of dry matter, respectively. Following hydrolysis of corn stover in a ratio of 100 g to 1000 L, approximately 2–10% by weight of xylan was converted to furfural [26], thus equivalent to 0.46–2.28 g/L depending on the elevation of the hydrolysis temperature. These results were in accordance with furfural contents obtained from xylose production from corn stover by means of diluted acid hydrolysis [27]. Based on this rationale that furfural may be in inhibitory in future optimization of corn stover substrate bioconversions, all 12 selected LAB isolates were evaluated for furfural tolerance at various concentrations ranging from 1 to 6 g/L. The results revealed that none of the LAB isolates were able to grow in medium supplemented with furfural at above 2 g/L. Growth and LA production of each LAB isolates in three tested carbon sources with and without supplementation of 2 g/L furfural were significantly different. Supplementation of furfural negatively affected LA production by decreasing LA production (Figure 2). Screening for furfural tolerance is an ability that can be applied for LA production as without hydrolysis or removal of the furfural the process can be a cost-effective method [28]. In this study the isolate WX1 was the least furfural sensitive lactic acid bacterium. It was active in the presence of 2 g/L furfural and retained the highest LA production efficiency with relative LA production of approximately 81.8%, 81.5% and 74.5% towards glucose, xylose and arabinose fermentations, respectively. Therefore, isolate WX1 was selected as the most potential isolate to be used for LA production in future corn stover-based process, while others could be alternative choices for LA production in combination with furfural hydrolysis process.

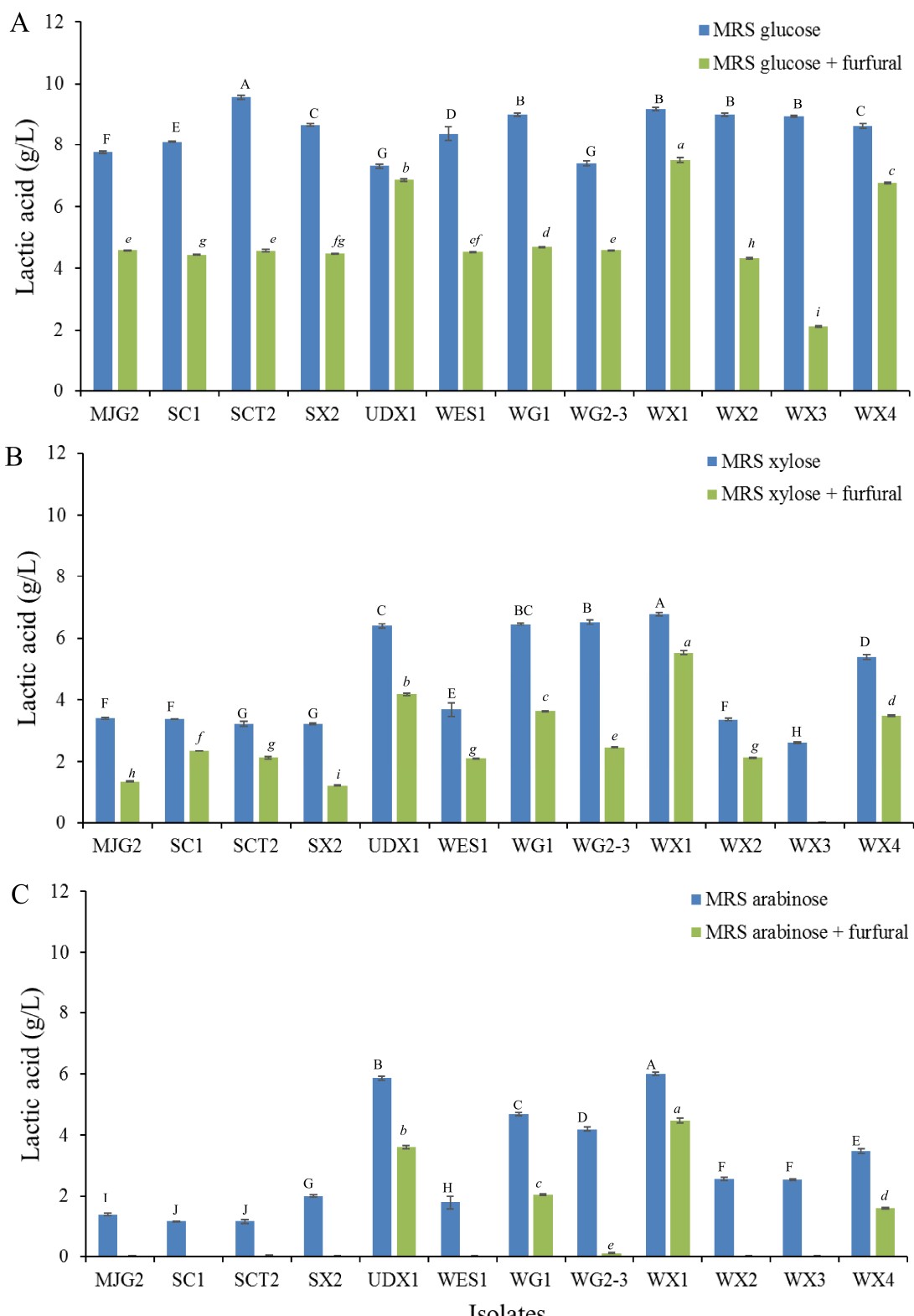

**Figure 2.** Lactic acid production of twelve selected bacterial isolates cultivated in MRS broth with 10 g/L glucose (**A**), xylose (**B**) and arabinose (**C**) supplemented with 2 g/L furfural at 37 °C for 48 h. Different capital letters (A–J) and small letters (a–i) indicate significant differences between the test groups of each media ($p < 0.05$).

### 3.4. Molecular Identification of LAB

The morphological study of 12 isolates found that 10 of the most acid-forming isolates including UDX1, WX4, WG2-3, WG1, WX1, MJG2, SCT2, SC1, W1ES and SX2 were Gram-positive cocci, while the last two isolates WX2 and WX3 were Gram-positive bacteria with irregular rod shape. The identification by 16S rRNA gene sequencing analyses revealed that all shared sequence similarity of higher than 99% to genera *Entercoccus* and *Weissella* which are in the order Lactobacillales. The full-length 16S rRNA gene sequences of 12 isolates have been deposited with GenBank under the accession no. MZ127632 - MZ127643. The isolates UDX1, WX4, WG2-3, WG1 and WX1 belonged to *Enterococcus mundtii* while isolates MJG2, SCT2, SC1 and W1ES were assigned as *Enterococcus faecalis*, and isolate SX2 was *Enterococcus hirae*. Two isolates WX2 and WX3 were *Weissella cibaria*. These results were in accordance with phylogenetic analysis results (Figure 3). *Enterococcus* and *Lactobacillus* were two main residents found in the midgut of mulberry silkworm, *Bombyx mori*. In addition, their relative abundance was varied depending on different kinds of silkworms which contributed to their physiological activities [29]. In the previous studies, *E. mundtii* QU25 was reported for its potential use for LA production from xylose. Although the fermentation was successful with different fermentation strategy and *E. mundtii* QU25 was the first LAB that was previously claimed as homofermentative LAB towards xylose, however a combination of xylose and glucose with furfural tolerance has not yet been reported as feasible for LA production [30,31]. Notably, this research study reveals potentially feasible solutions to the challenge of using *E. mundtii* WX1 as LA producer in future optimization from chemical treated lignocellulosic materials with furfural tolerance.

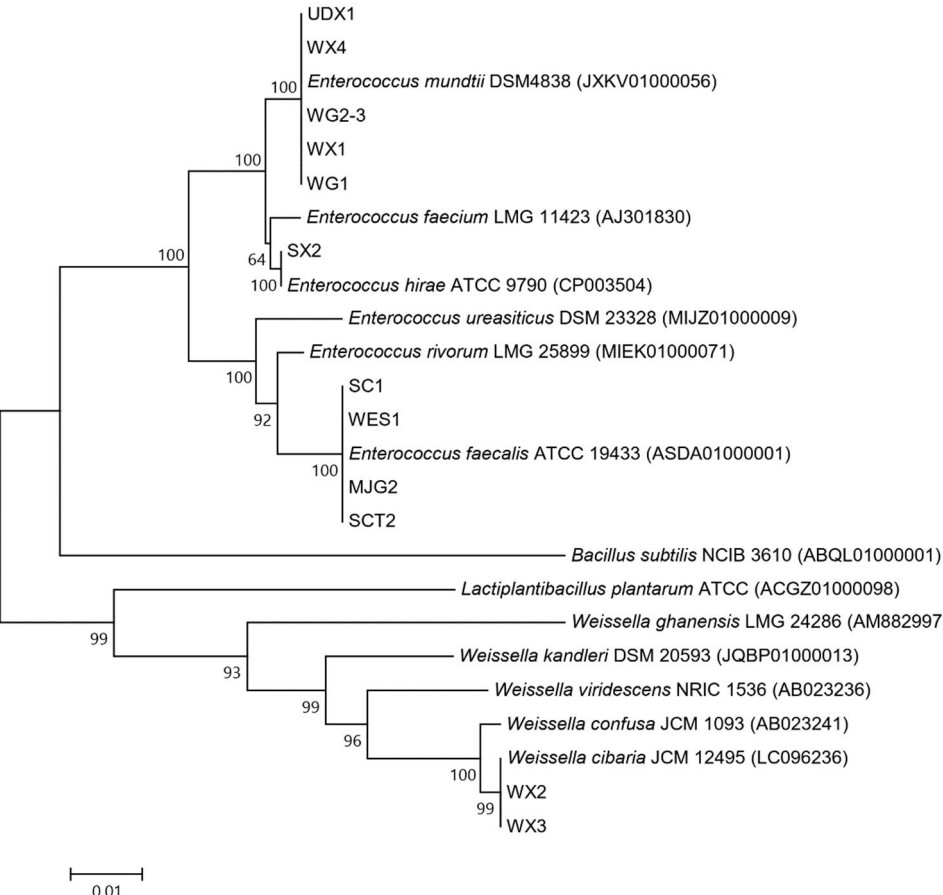

**Figure 3.** Phylogenetic tree of the isolate WX1 and other 11 isolates based on the 16S rRNA gene sequence analysis.

### 3.5. Effect of Glucose and Xylose Concentrations on L-LA Production from E. mundtii WX1

Chemical lignocellulosic hydrolysates typically contain glucose and xylose as the main fermentable sugars and their ratios are varied depending on material types and sources [12]. Aligning this future use, in this study six ratios of sugar mixture were prepared for screening LA production. The results are shown in Table 2. *E. mundtii* WX1 selectively consumed glucose and rapidly produced LA within 12 h of cultivation (data not shown), while xylose was fermented later. It indicated that at high concentration of glucose rapid glucose consumption resulted but decreased xylose utilization. This phenomenon could be found in a medium containing glucose as high as 50% of sugar mixture for example G15X5 and G10X10 where xylose was utilized after initially glucose had been almost or completely consumed by *E. mundtii*. The phenomenon is called glucose repression. Carbon utilization by lactic acid bacteria is regulated by carbon catabolite repression (CCR). Previously, glucose repression has been explained in *L. pentosus* as its xylose utilization is controlled by *xylAB* operon, encoding D-xylose isomerase and D-xylulose kinase which are required for xylose fermentation but can be repressed initially by the presence of glucose [32]. The same CCR was also found in *L. brevis* and *L. bucheri* [33] as well as *E. mundtiii* QU25 [34]. Here, coculture using two microorganisms that are complimentary to utilizing each sugar in terms carbon preference is an alternative strategy to overcome the main drawback. The first lactic acid bacterium should play a crucial role in rapid utilization of glucose in order to reduce the CCR effect which in turn then provides succession conditions for xylose utilization by *E. mundtii* WX1.

**Table 2.** The remaining glucose and xylose and L-lactic acid produced by *E. mundtii* WX1 using various glucose-xylose mixtures in MRS broth under static incubation at 37 °C for 48 h.

| Treatment | Time (h) | Glucose (g/L) | Xylose (g/L) | Total Sugar (g/L) | L-Lactic Acid (g/L) | Yield (g/g) |
|---|---|---|---|---|---|---|
| G20 | 0 | 20.00 ± 0.01 | 0 | 20.00 ± 0.02 | 0 | - |
| | 24 | 5.51 ± 0.02 | 0 | 5.51 ± 0.02 | 14.02 ± 0.02 | 0.97 |
| | 48 | 1.45 ± 0.01 | 0 | 1.45 ± 0.02 | 17.94 ± 0.02 | 0.97 |
| G15X5 | 0 | 15.00 ± 0.05 | 5.00 ± 0.02 | 20.00 ± 0.02 | 0 | - |
| | 24 | 0 | 4.25 ± 0.02 | 4.25 ± 0.01 | 10.89 ± 0.01 | 0.69 |
| | 48 | 0 | 2.82 ± 0.03 | 2.82 ± 0.03 | 11.37 ± 0.02 | 0.66 |
| G10X10 | 0 | 10.00 ± 0.02 | 10.00 ± 0.02 | 20.00 ± 0.02 | 0 | - |
| | 24 | 0.14 ± 0.01 | 6.93 ± 0.01 | 7.07 ± 0.01 | 8.33 ± 0.02 | 0.64 |
| | 48 | 0 | 2.39 ± 0.01 | 2.39 ± 0.01 | 10.89 ± 0.02 | 0.61 |
| G5X15 | 0 | 5.00 ± 0.02 | 15.00 ± 0.01 | 20.00 ± 0.01 | 0 | - |
| | 24 | 0 | 12.17 ± 0.01 | 12.17 ± 0.01 | 7.37 ± 0.01 | 0.94 |
| | 48 | 0 | 7.99 ± 0.01 | 7.99 ± 0.01 | 9.77 ± 0.02 | 0.81 |
| X20 | 0 | 0 | 20.00 ± 0.01 | 20.00 ± 0.01 | 0 | - |
| | 24 | 0 | 9.48 ± 0.02 | 9.48 ± 0.02 | 10.09 ± 0.02 | 0.96 |
| | 48 | 0 | 5.66 ± 0.02 | 5.66 ± 0.02 | 13.05 ± 0.03 | 0.91 |

Note: Yield (g/g) = lactic acid formation (g)/total sugar consumption (g). The value below the limit of detection by enzymatic assay kit was marked as "0".

### 3.6. L-LA Production from Mixed Glucose and Xylose by Coculture Fermentation

According to the previous reports on the ratio of glucose and xylose in corn stover after pretreatment and hydrolysate generation was found to be approximately 60% glucose and 40% xylose [22,35], and therefore the mixed sugar composition of glucose (C6) and xylose (C5) in a ratio of 6:4 was selected as a simplified model for screening L-LA fermentation in the isolates of this study. The results showed that cofermentation of *E. mundtii* WX1 and *L. rhamnosus* SCJ9 can utilize mixed glucose and xylose both in the presence and absence of furfural (Figure 4). In the presence of 2 g/L furfural, L-LA titer, production efficiency and productivity for cofermentation were 19.82 g/L, 0.66 g/g and 0.41 g/L/h, respectively,

whereas the furfural absence condition achieved 23.32 g/L, 0.77 g/g and 0.48 g/L/h, respectively at 48 h. From this result, it could be suggested that fermentation of mixed sugar using coculture led to a higher resultant L-LA titer when compared to the use of single culture. Similar results were also reported by Taniguchi, et al. [36] when a mixture of 50 g/L of xylose and 100 g/L of glucose was used as the carbon source in the cultivation of *E. casseliflavus*, where glucose was converted to LA in the early phase of the cultivation, but xylose was hardly consumed. The cocultivation of *E. casseliflavus* and *L. casei* with two-stage inoculation resulted in complete consumption of 50 g/L of xylose and 100 g/L of glucose and achieved 95 g/L of LA with a high optical purity of 96% at 192 h.

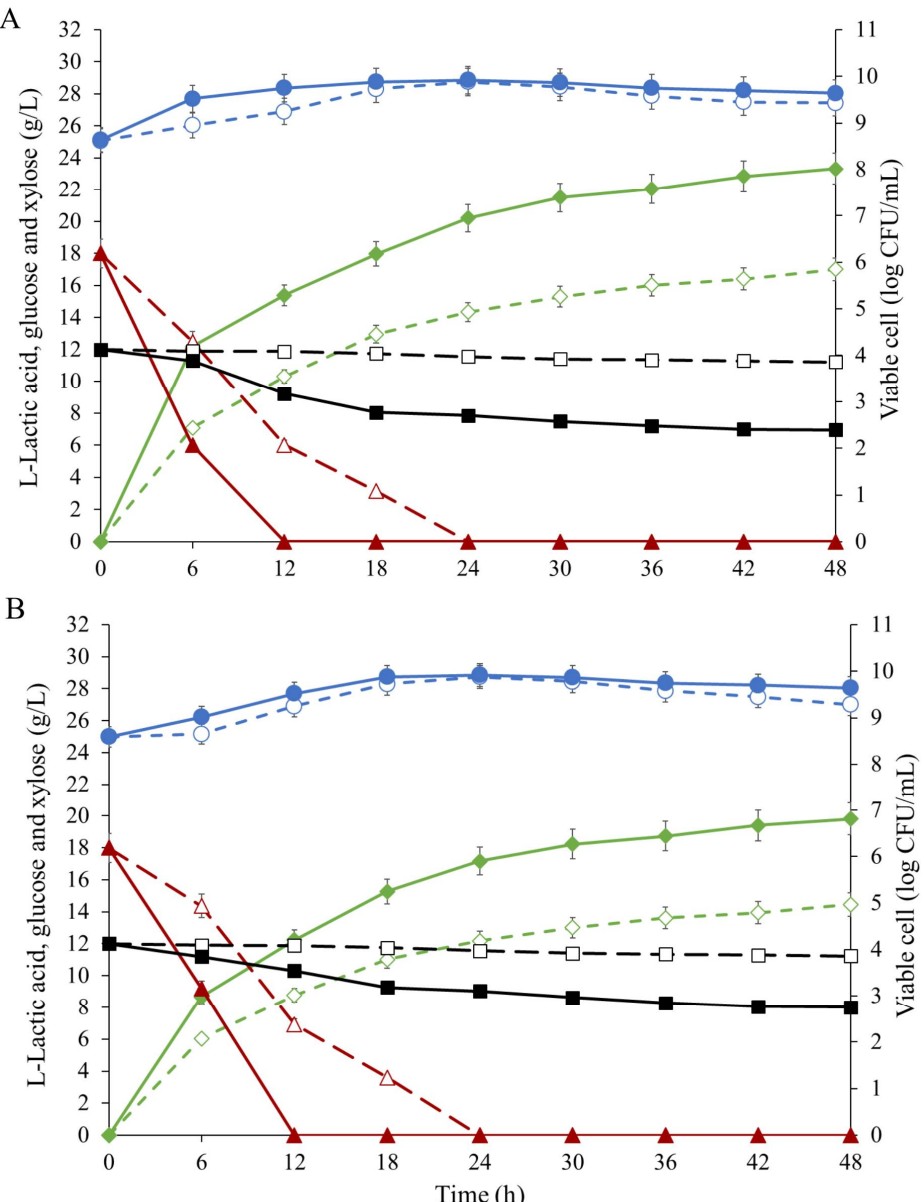

**Figure 4.** Comparison of lactic acid production by coculture of *E. mundtii* WX1 and *L. rhamnosus* SCJ9 (solid line) and single-culture of *E. mundtii* WX1 (dashed line) in MRS with 30 g/L mixed glucose-xylose carbon source without furfural (**A**) and in the presence of 2 g/L furfural (**B**) at 37 °C for 48 h. Symbols; (— ◆ —, - - ◇ - -) lactic acid, (— ▲ —, - - △ - -) glucose, (— ■ —, - -□- -) xylose and (— ● —, - -○- -) viable cell.

### 3.7. Medium Optimization for L-LA Production by Coculture Using Experimental Designs

PBD was used to find the optimum significant compositions for L-LA production by *E. mundtii* WX1 and *L. rhamnosus* SCJ9 from nine medium compositions based on MRS medium including 30 g/L carbon source (60% glucose: 40% xylose), YE, peptone, beef extract, Tween 80, Tri-ammonium citrate, sodium acetate, $MgSO_4 \cdot 7H_2O$, $K_2HPO_4$ and $MnSO_4 \cdot H_2O$ (Supplementary Materials, Table S2). Using the Design Expert software, the data were analyzed by the linear regression in order to fit with the first-order model and found that the model was highly significant for *p*-value < 0.05 with $R^2$ = 0.9661 and adjusted $R^2$ = 0.8897 (Table 3). The analysis of variance (ANOVA) indicated the four most important variables which included Tween 80, YE, $MnSO_4 \cdot H_2O$ and sodium acetate. Tween 80, YE and $MnSO_4 \cdot H_2O$ influenced with positive coefficient (+), whereas the negatively influencing factor was sodium acetate, which has inverse response on LA production. Tween 80 has been accepted for its function as the emulsifier to assist the dispersion of nutrients in a variety of bacterial cultivation and it is one important component in MRS medium [37]. Several studies reported that Tween 80 strongly affected LA production and cell growth [38–40]. The rich organic nitrogen source such as YE, beef extract, meat extract and peptone consisted of the important purine and pyrimidine bases and B vitamins, which are essential components for the growth of LAB and YE has been used in most of the LA production studies as the organic nitrogen supplement [41–43]. The significant influence by $Mn^{2+}$ ion might be caused from the involvement of $Mn^{+2}$ ion in L-lactate dehydrogenase activity as described by Narayanan, et al. [1]. PBD offers an efficient route to optimize large number of variables and identifies the most significant variables. The design has been successfully applied in medium optimization by various previous studies [44–47].

**Table 3.** The analysis of variance (ANOVA) from the PBD experimental results.

| Variable Code | Medium Components | Coefficient Estimate | Sum of Squares | df | F-Value | p-Value (Prob > F) |
|---|---|---|---|---|---|---|
| | Model/Intercept | 12.22 | 105.03 | 10 | 12.65 | 0.0132 |
| A | Tween 80 | 1.34 | 21.55 | 1 | 23.36 | 0.0084 |
| C | Peptone | −0.40 | 3.05 | 1 | 3.31 | 0.1431 |
| D | Beef extract | 0.024 | 0.011 | 1 | 0.012 | 0.9197 |
| E | $MgSO_4 \cdot 7H_2O$ | −0.34 | 1.41 | 1 | 1.53 | 0.2840 |
| F | $MnSO_4 \cdot H_2O$ | 1.71 | 34.91 | 1 | 37.84 | 0.0035 |
| G | $K_2HPO_4$ | 0.26 | 0.83 | 1 | 0.90 | 0.3958 |
| H | Tri-ammonium citrate | 0.11 | 0.15 | 1 | 0.16 | 0.7106 |
| K | Sodium acetate | −0.97 | 11.20 | 1 | 12.14 | 0.0253 |
| L | YE | 1.63 | 31.91 | 1 | 34.58 | 0.0042 |

$R^2$ = 0.9661; Adj $R^2$ = 0.8897.

CCD was employed to optimize the concentrations of these important three variables for maximum L-LA production. The design matrix of the variables in actual values is presented in Table 4 with the response results. By fitting the experimental L-LA with the least square linear regression models, the simulated second order equation was obtained as Equation (4):

$$Y = 2.94 + 0.73X_1 + 15.43X_2 + 3.18X_3 + 0.1X_1X_2 + 0.05X_1X_3 + 3.26X_2X_3 - 0.02X_1{}^2 - 7.50X_2{}^2 - 3.57X_3{}^2 \tag{4}$$

where Y is the predicted response L-LA, $X_1$, $X_2$ and $X_3$ were YE, Tween 80 and $MnSO_4 \cdot H_2O$, respectively.

**Table 4.** Experimental designs and the experimental results of L-LA production responding to the CCD.

| Run | $X_1$: YE (g/L) | $X_2$: Tween 80 (g/L) | $X_3$: $MnSO_4 \cdot H_2O$ (g/L) | Lactic Acid (g/L) |
|---|---|---|---|---|
| 1 | 9 | 1 | 0.6 | 20.02 |
| 2 | 31 | 1 | 0.6 | 19.45 |
| 3 | 9 | 2 | 0.6 | 16.29 |
| 4 | 31 | 2 | 0.6 | 17.20 |
| 5 | 9 | 1 | 2.2 | 14.87 |
| 6 | 31 | 1 | 2.2 | 15.48 |
| 7 | 9 | 2 | 2.2 | 15.57 |
| 8 | 31 | 2 | 2.2 | 19.24 |
| 9 | 1.5 | 1.5 | 1.4 | 14.35 |
| 10 | 38.5 | 1.5 | 1.4 | 15.30 |
| 11 | 20 | 0.66 | 1.4 | 17.98 |
| 12 | 20 | 2.34 | 1.4 | 16.99 |
| 13 | 20 | 1.5 | 0.05 | 16.63 |
| 14 | 20 | 1.5 | 2.75 | 16.02 |
| 15 | 20 | 1.5 | 1.4 | 23.97 |
| 16 | 20 | 1.5 | 1.4 | 22.95 |
| 17 | 20 | 1.5 | 1.4 | 24.08 |
| 18 | 20 | 1.5 | 1.4 | 24.10 |
| 19 | 20 | 1.5 | 1.4 | 24.10 |
| 20 | 20 | 1.5 | 1.4 | 22.97 |

A summary of the analysis of variance (ANOVA) for response surface quadratic model is presented in Table 5. The model was highly significant at *p*-value 0.0001 with $R^2$ value 0.9519 which was in the agreement with the adjusted $R^2$ value (adjusted $R^2$ = 0.9086) and indicated good fit of the model. The result inferred that Tween 80 was the significant variable for L-LA production by coculture *E. mundtii* WX1 and *L. rhamnosus* SCJ9 with *p*-value 0.011. The significant effects were also observed from the interaction of the two variables of the model corresponding to the convex shape of the 3D contour response surface plot (Figure 3). The predicted maximum level of L-LA was 23.72 g/L with 20.61 g/L YE, 1.44 g/L Tween 80, 1.27 g/L $MnSO_4 \cdot H_2O$ and with 30 g/L carbon source. L-LA production model was validated within 48 h growth and the highest L-LA of 23.59 ± 0.5 g/L was obtained from optimized medium. Even the difference of L-LA obtained from the optimized medium is not large when compared to that from the MRS, but the optimized medium could reduce the high-cost nitrogen sources such as beef extract and peptone, with levels dropping from 10 g/L to 2.5 g/L. Furthermore, other significant negatively influencing nutritional factors such as sodium acetate were omitted from the optimized medium. The overall main purpose of this study is the advancement of long term application strategy for conversion of corn stover or other lignocellulose substrates to L-LA. Therefore, the initial concentration of simple mixed sugar substrates was fixed at 30 g/L, which is equivalent to the fermentable sugar expected to be achieved from 100 g dried corn stover or other lignocellulosic substrates [22]. Azaizeh, et al. [48] recommended that 10% dry matter (DM) was suitable for lignocellulose biomass pretreatment and hydrolysis and DM higher than 15% may cause a technical problem due to high viscosity [49].

**Table 5.** ANOVA for response surface quadratic model of lactic acid production.

| Source | Coefficient Estimate | df | MS | F Value | *p*-Value (Prob > F) |
|---|---|---|---|---|---|
| Model | 23.65 | 9 | 25.55 | 21.98 | <0.0001 |
| $X_1$ | 0.45 | 1 | 2.83 | 2.43 | 0.1498 |
| $X_2$ | −0.23 | 1 | 0.73 | 0.63 | 0.4457 |
| $X_3$ | −0.65 | 1 | 5.68 | 4.89 | 0.0515 |
| $X_1 X_2$ | 0.57 | 1 | 2.57 | 2.21 | 0.1682 |
| $X_1 X_3$ | 0.49 | 1 | 1.95 | 1.68 | 0.2240 |
| $X_2 X_3$ | 1.31 | 1 | 13.64 | 11.74 | 0.0065 |
| $X_1{}^2$ | −2.82 | 1 | 114.43 | 98.44 | <0.0001 |
| $X_2{}^2$ | −1.88 | 1 | 50.78 | 43.69 | <0.0001 |
| $X_3{}^2$ | −2.29 | 1 | 75.40 | 64.87 | <0.0001 |
| Residual | | 10 | 1.16 | | |
| Lack of fit | | 5 | 2.00 | 6.12 | 0.0343 |
| Pure error | | 5 | 0.33 | | |

$R^2$ = 0.9519; Adj $R^2$ = 0.9086.

### *3.8. Comparative Study on L-LA Production by the Optimized Medium*

Production of L-LA by coculture of *E. mundtii* WX1 and *L. rhamnosus* SCJ9 using the optimized medium containing 30 g/L mixed glucose and xylose (6:4) was compared to the use of MRS broth inoculated with coculture and single culture *E. mundtii* WX1 (Table 6). Considering the level of L-LA production, approximately 80% of the final L-LA accumulation was achieved by 24 h cultivation in all cultivation strategies and the use of coculture both in optimized medium and MRS clearly improved the bioconversion of mixed glucose-xylose to L-LA in comparison to the single culture as it showed higher L-LA yield than *E. mundtii* WX1 single culture at levels of approximately 1.53 and 1.28 times, respectively. The results from this experiment indicated the possibility of using the coculture of *E. mundtii* WX1 and *L. rhamnosus* SCJ9 in future bioconversion strategies for lignocellulose particularly corn stover to L-LA, a basic building process for polylactide synthesis. However, the research to determine more suitable fermentation processes and strategies, along with lower cost medium components are essential.

**Table 6.** Comparison of lactic acid titer, production efficiency and productivity of coculture *E. mundtii* WX1 and *L. rhamnosus* SCJ9 cultivated in optimized medium and MRS with 30 g/L mixed glucose-xylose carbon source in the presence of 2 g/L furfural at 37 °C in comparing to the single culture.

| Parameter | Fermentation Strategy | | | | | |
|---|---|---|---|---|---|---|
| | WX1 MRS | | WX1 and SCJ9 MRS | | WX1 and SCJ9 Optimized Medium | |
| | 24 h | 48 h | 24 h | 48 h | 24 h | 48 h |
| Lactic acid (g/L) | 12.07 ± 0.5 | 15.51 ± 0.3 | 17.56 ± 0.4 | 19.82 ± 0.1 | 20.88 ± 0.8 | 23.59 ± 0.5 |
| Production efficiency (g/g) | 0.49 ± 0.01 | 0.48 ± 0.01 | 0.59 ± 0.01 | 0.66 ± 0.04 | 0.70 ± 0.02 | 0.76 ± 0.01 |
| Productivity (g/L/h) | 0.50 ± 0.02 | 0.30 ± 0.07 | 0.73 ± 0.01 | 0.41 ± 0.09 | 0.87 ± 0.03 | 0.48 ± 0.01 |

## 4. Conclusions

*E. mundtii* WX1 was selected from acid-producing bacteria from Eri silkworm midgut with a capability for L-LA production when grown on glucose or xylose and arabinose. However, *E. mundtii* WX1 was incapable of utilizing xylose in the presence of glucose higher than 50% in mixed combination of hexose-pentose sugars. This research advanced a solution by using coculture of *E. mundtii* WX1 and *L. rhamnosus* SCJ9 and achieved L-LA levels 1.28 times higher than those obtained by single culture *E. mundtii* WX1. The optimal medium component for L-LA production for *E. mundtii* WX1 and *L. rhamnosus* SCJ9 coculture based on MRS was improved by PBD and CCD experimental designs,

which enhanced LA production by 1.53 times. This is the first report describing the use of LAB from a novel source such as Eri silkworm midgut and advancing the excellent strategic possibility to use the coculture of *E. mundtii* WX1 and *L. rhamnosus* SCJ9 in a future bioconversion process of lignocellulose, particularly corn stover to L-LA.

**Supplementary Materials:** The following are available online at https://www.mdpi.com/article/10.3390/fermentation7020095/s1, Table S1: Medium components and their levels for PBD experimental design, Table S2: l-Lactic acid production by coculture of *E. mundtii* WX1 and *L. rhamnosus* SCJ9 responding to the designed matrix of Plackett-Burman design under static condition at 37 °C for 48 h.

**Author Contributions:** Conceptualization, A.K. (Augchararat Klongklaew) and C.K.; methodology and formal analysis, A.K. (Augchararat Klongklaew), P.P. and K.U.; investigation, A.K. (Augchararat Klongklaew), A.K. (Apinun Kanpiengjai), K.U., P.W., P.P., and C.K.; writing-original draft preparation, A.K. (Augchararat Klongklaew), K.U., and C.K.; writing-review and editing, A.K. (Augchararat Klongklaew), A.K. (Apinun Kanpiengjai), P.W., K.U., K.S. and C.K.; supervision, C.K. All authors have read and agreed to the published version of the manuscript.

**Funding:** This work was financially supported by Chiang Mai University's 50th anniversary scholarship program.

**Institutional Review Board Statement:** Not applicable.

**Informed Consent Statement:** Not applicable.

**Data Availability Statement:** Not applicable.

**Acknowledgments:** The authors acknowledge Chiang Mai University for Chiang Mai University's 50th anniversary scholarship and also the Faculty of Agro-Industry, Chiang Mai University for the research facilities.

**Conflicts of Interest:** The authors declare no conflict of interest.

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
