# Peer review of "Improvement of Enantiomeric l-Lactic Acid Production from Mixed Hexose-Pentose Sugars by Coculture of Enterococcus mundtii WX1 and Lactobacillus rhamnosus SCJ9"

_fermentation, doi:10.3390/fermentation7020095_

Round 1
Reviewer 1 Report
The manuscript submitted for evaluation describes the use of lactic acid bacteria isolated from Eri silkworm for the production of lactic acid. The studies included the selection of microorganisms and optimization of the biosynthesis of components in the medium.
The presented manuscript lacks information on how the residual sugar after fermentation was determined. What methods were used to determine them? What were the parameters of the lactic acid determination method? What were the LOD values Table 2. What does "0" mean for lactic acid content and yield? Does that mean these values are below the LOD?
Reviewer 2 Report
Line 43: ”…the increase in demand for…” Are both isomeric forms equally sought after?
Line 56: “…LA production” At what levels?
Line 61: “its” instead of “their”
Line 67: “LAB”, Lactic acid bacteria?
Line 230-234: seems to be a repetition of the text above
Figure 1: “from” instead of “form”, what do the letters gh, hi etc. stand for?
Line 253-254: “…sugars derived from lignocellulose…” Did you use pure sugars or hydrolysates?
Line 259: “…3.2 and 1.15 g/L…” Maybe you should compare with the levels produced by WX1?
Line 274: “…chemical hydrolysis generated…” Did you do this chemical hydrolysis? Cannot find it in the method section.
Line 283: “…all were not able…” were some able, or none?
Line 290: “least” instead of “lowest”
Line 299: “10 of the most” instead of “10 most of”
Line 317: “for LA” instead of “LA for”
Line 352: How/why was L. rhamnosus SCJ9 chosen?
Reviewer 3 Report
The authors in the manuscript “Improvement of Enantiomeric L-Lactic Acid Production from Mixed Hexose-Pentose Sugars by Co-Culture of Enterococcus mundtii WX1 and Lactobacillus rhamnosus SCJ9” described the isolation of LAB and the L-Lactic acid production using co-cultures. The topic is interesting, and the methods, results, and discussion sections were well conducted.
My main concern is the sugars used. In the introduction the authors addressed the lignocellulose material (L60-71), however, I consider that they should carry out experiments with this material, since as they describe it is the most abundant biomass in the world. MRS media is a commercial broth with 20g/L of glucose, also contains other specific components, however, using other minimal media can reduce the cost and the material consume. The lignocellulose supplementation can be an alternative to see the real effect of the co-culture experiments.
In each graph, what the letters mean? What type of statistical analysis was carried out? Authors must describe what software and what analysis was performed for all experiments.
L110. Are they added 10 g / L of xylose plus 20 g / L of glucose contained in the MRS medium?
L382. Delete the sentence “the selective medium for LAB”.
L436. Put the word “expe riment” together
